# Exposure to Sulfur Hexafluoride Influences Viability in Cell Transplant Suspensions

**DOI:** 10.3390/biotech14040086

**Published:** 2025-10-31

**Authors:** Laura Martínez-Alarcón, Sergio Liarte, Juana M. Abellaneda, Juan J. Quereda, Livia Mendonça, Antonio Muñoz, Pablo Ramírez, Guillermo Ramis

**Affiliations:** 1Servicio de Cirugía, Hospital Clínico Universitario Virgen de la Arrixaca, 30120 Murcia, Spain; lma5@um.es (L.M.-A.); pablo.ramirez@carm.es (P.R.); 2Instituto Murciano de Investigación Biosanitaria (IMIB), El Palmar, 30120 Murcia, Spain; antmunoz@um.es (A.M.); guiramis@um.es (G.R.); 3Health Sciences Faculty, Universidad Católica San Antonio de Murcia (UCAM), Av. de los Jerónimos, 135, Guadalupe de Maciascoque, 30107 Murcia, Spain; 4Grupo de Investigación Cría y Salud Animal, Universidad de Murcia, 30100 Murcia, Spain; juanamaria.abellaneda@um.es; 5Departamento Producción y Sanidad Animal, Salud Pública Veterinaria y Ciencia y Tecnología de los Alimentos, Facultad de Veterinaria, Universidad Cardenal Herrera-CEU, 46115 Valencia, Spain; juan.quereda@uchceu.es; 6Escola de Vetérinaria, Universidade Federal de Goiás, Goiánia 74001-970, Brazil; liviapascoal@ufg.br; 7Departamento de Producción Animal, Universidad de Murcia, 30100 Murcia, Spain

**Keywords:** ultrasonography contrast, RTCA, cytotoxicity, cell therapy, transplantation

## Abstract

Cell transplantation is often performed with ultrasonographic guidance for accurate delivery through injection. In such procedures, using ultrasonographic contrast greatly improves target delivery. However, accumulating evidence suggests that exposure to such contrast agents may have negative effects on transplanted cells. No study so far has researched this issue. Stabilized sulfur hexafluoride (SF6) microbubbles are a widely used sonographic contrast agent. Skin hCD55 porcine transgenic fibroblasts and mesenchymal stem cells from human bone marrow (hMSCs) were exposed in vitro to SF6 in concentrations ranging from 1.54 µM to 308 µM. The effects on viability and cell growth were registered using an impedance-based label-free Real-Time Cell Analyzer (RTCA). Data was recorded every 15 min for 50 h of total study time. Both cell lines behave distinctly when exposed to SF6. Porcine fibroblast growth showed relevant alterations only when exposed to higher concentrations. In contrast, hMSCs showed progressive growth decrease in relation to SF6 concentration. Taken together, while SF6-based contrast agents pose no threat to patient safety, our results indicate that exposure of suspended stem cells to the contrast agent could affect the effective dose administered in cell therapy procedures. This prompts specific cell lineage testing, adjusting methods and properly compensating for cell loss, with a potential impact on procedural cost and success rates.

## 1. Introduction

Cell therapy procedures are becoming an increasingly suitable resource for the treatment and management of previously orphan conditions, including hematological and degenerative ailments. At the clinical level, eco-guided injection techniques are often applied, especially when delivering cells to areas that are difficult to reach and/or localized compartments. In such cases, operators can apply ultrasound contrast inoculation along with cell suspensions in order to verify targeted delivery.

Among the available agents, sulfur hexafluoride (SF_6_) is a gas belonging to the latest generation of sonographic contrast agents. Such formulations can be regarded as among the first examples of successful nanotechnology applications in the medical field, having contributed to advances in contrast-enhanced ultrasonography procedures, which are increasingly common in clinical practice. SF_6_ gas is stabilized into microbubbles by means of proprietary phospholipid blends that organize into microscopic micelles. SF_6_ formulations have been shown to be safe for patients and are especially useful in cardiology and liver imaging [1,2,3,4]. Notably, compared to alternative contrast reagents used for electromagnetic imaging techniques, i.e., a CT scan or magnetic resonance imaging, SF_6_ is regarded as being free of nephrotoxicity [5]. However, accumulated evidence shows that damage to tissues exposed to SF_6_ formulations is real, especially when used in conjunction with high-energy ultrasound waves [6,7,8]. In this sense, SF_6_ microbubbles have been shown to cavitate when exposed to sound waves, releasing energy to their surroundings. Such releases have also been shown to be able to generate infinitesimal perforations in cell membranes, altering permeability. In fact, this effect is being considered for optimizing the delivery of drugs or even being used to dismantle microvessels within tumor tissues [6,7,8]. While SF_6_ formulas have been cleared as safe and approved for indicated clinical imaging uses [9], in the case of eco-guided cell injection techniques, to our knowledge, no data is available regarding any potential detrimental impact derived from their direct interaction with injected cells. It is important to note that in the indicated procedures, i.e., the imaging of vascular cavities, the biological structures involved do endure quick and brief exposures to SF_6_ microbubbles, mainly due to vascular flow and thus mechanical clearance. However, in specific cell therapy applications, suspensions carrying cells and ultrasonographic contrast can persist for extended periods, at least for the duration of the procedure, and presumably until complete mechanical clearance and/or chemical degradation of the agent at the delivery location.

Based on the evidence above, considerable uncertainty arises, as there is a possibility that the direct interaction between cells and SF_6_ microbubbles may have detrimental effects, potentially affecting procedure performance. Consequently, this study aims to evaluate the potential existence of detrimental effects, in terms of cytotoxicity, for long-term exposure to the SF_6_ sonographic contrast agent SonoVue^®^ on stem cells, resorting to lineages previously used in our lab for eco-guided in utero cell injection procedures [10], namely human mesenchymal stem cells (hMSCs) and transgenic porcine fibroblast.

## 2. Materials and Methods

### 2.1. Cells, Origin, and Ethical Assessment

Human mesenchymal stem cells (*hMSCs*) were obtained from the bone marrow of one healthy volunteer donor. The subject was provided with a full description of the study and provided informed consent. Details on the experimental procedure and setup involving human cells were waived by the Ethics Committees of the Hospital Virgen de la Arrixaca. Transgenic porcine fibroblast cells were obtained as primary cultures from fresh skin samples from two human-CD55 (*hCD55*) transgenic Landrace–Large white pigs (*Sus scrofa domestica*; Immutran, UK). Pig husbandry was carried out in compliance with the European Convention for the Protection of Vertebrate Animals used for Experimental Purposes (R.D. 324/2000) [11]. Details on the experimental procedure and setup involving pigs were approved by the Ethics Committees of the University of Murcia.

### 2.2. Human Mesenchymal Stem Cell Isolation, Culture, and Expansion

HMSCs were isolated from adult human bone marrow homogenates by means of density gradient centrifugation (1.077 g/cm^3^) at 400× *g* for 30 min at room temperature using Histopaque-1077 (Sigma-Aldrich, Burlington, MA, USA). The interphase containing the mononuclear cells was washed three times using PBS (Invitrogen, Waltham, MA, USA), followed by centrifugation at 250× *g* for 10 min. The resulting cell pellet was resuspended in 1 mL of Dulbecco’s modified Eagle medium (DMEM) containing 10% fetal bovine serum, 1% L-glutamine (2 mmol/L), and 1% penicillin/streptomycin (all reagents obtained from Gibco [Thermo-Fisher Scientific, Waltham, MA, USA]). Cell count and viability were determined using a Neubauer chamber along with the trypan blue staining exclusion method. Viable cells were seeded at a density of 1 × 10^6^ cells per cm^2^ in 25 cm^2^ flasks and placed in a humidified incubator at 37 °C with 5% CO_2_ for 72 h. Upon reaching 90% confluence, cells were washed 3 times with PBS (Invitrogen, Waltham, MA, USA) at room temperature and detached from flasks using trypsin–EDTA (Invitrogen, Waltham, MA, USA) for 5 min at 37 °C. Detached cells were seeded and sub-cultured in fresh 25 cm^2^ flasks. Subsequent passages were also performed upon 90% confluence. Cells in all experiments were taken from passages 5 to 10.

### 2.3. Transgenic Porcine Fibroblast Isolation, Culture, and Expansion

Transgenic porcine primary fibroblasts were isolated from fresh skin samples of the inner side of the fore. Primary culture cells were obtained by brief enzymatic digestion with collagenase (Sigma-Aldrich, Burlington, MA, USA), then were grown and expanded in culture flasks using DMEM containing 10% fetal bovine serum, 1% L-glutamine, and 1% penicillin/streptomycin (all from Gibco [Thermo-Fisher Scientific, Waltham, MA, USA]). Cells were incubated in a humidified atmosphere at 37 °C and 5% CO_2_. Upon reaching 90% confluence, cells were washed 3 times with PBS (Invitrogen, USA) at room temperature and detached from flasks using trypsin (Invitrogen, Waltham, MA, USA) for 5 min at 37 °C. Detached cells were seeded and sub-cultured in fresh 25 cm^2^ flasks. Subsequent passages were also performed upon 90% confluence. Cells in all experiments were taken from passages 5 to 10.

### 2.4. Contrast Agent

SonoVue^®^ is a second-generation sonographic contrast that consists of a suspension of phospholipid-stabilized sulfur hexafluoride (SF_6_) microbubbles (Bracco SpA, Italy). The mean diameter of these microbubbles is 2.5 μm (range of 0.7 to 10 μm). The mechanical index constitutes a unitless ultrasound metric, which is used as an indication of potential cavitation bio-effects. In this regard, SF_6_ microbubbles are not destroyed when a low-mechanical-index sonographic technique is applied (range 0.07–0.2). In a previous in vivo setting, this allowed for real-time scanning during in utero cell injection [10]. To test potential cytotoxicity, in this study, the ultrasound contrast agent was reconstituted in saline solution according to the manufacturer’s protocol and then further diluted by mixing with the culture medium present in the assay well to obtain a final concentration that ranged from 2 to 5 × 10^8^ microbubbles per mL, conforming to a range from 1.54 to 308 µM.

### 2.5. Real-Time Cytotoxicity Assay

The real-time cytotoxicity assay (RTCA) was performed under validated conditions [12,13,14,15] by means of the xCELLigence^®^ SP RTCA system (ACEA Bio, San Diego, CA, USA). The RTCA is based on the assessment of impedance variations, depending on culture confluence, by using a microelectronic 96-well culture plate (E-plate; ACEA Bio, San Diego, CA, USA). From the impedance values, the system calculates a dimensionless parameter known as the cellular index (CI), which enables us to compare the overall cell integrity between samples. For this study, either hMSCs or porcine fibroblasts were seeded into E-plates at a density of 7.5 × 10^3^ cells in 150 µL of supplemented cell culture medium per well. Cells were allowed to grow for 10 h in order to obtain a CI higher than 0.5, as recommended by the manufacturer. For challenge, different culture medium volumes were exchanged with saline SF_6_ suspensions up to a total assay volume of 200 µL per well, with final corresponding SF_6_ concentrations ranging from 308 µM to 1.54 μM. In sham control assays, the well volume was adjusted with the same saline quantity used to obtain the final SF_6_ suspension with the highest concentration. Both hMSC and porcine fibroblast assays were performed on three independent occasions, with duplicates for each condition. After challenge, the impedance was registered automatically every 15 min for a total of 50 h. For analysis purposes, individual time point CI data for each well were normalized by the individual CI values obtained right before challenge. The minimum CI (CI_min_), corresponding to the maximum cytotoxicity for each condition, was calculated as the lowest normalized CI observed after challenge throughout the experiment; in addition, the time invested to reach CI_min_ in hours was recorded (TCI_min_). To estimate the percentage CI decrease (CI∇_%_) over time for each condition, the calculations were as follows: CI∇_%_ = (1 − CI_min_) × 100; the CI value was 1 at the initial normalization time point. In addition, for simplified assessment, the area under the curve (AUC) and the CI_min_ vs. concentration were calculated.

### 2.6. Statistics and Correlation Analysis

RTCA xCELLigence^®^ system software v.1.2 was used to calculate both the correlation index and *p*-values for the AUC and CI_min_ for the time period vs. concentration. The differences in CI_min_ for a time period vs. concentration were assessed by linear regression, and the percentage CI decrease (CI∇_%_) vs. the number of cells alive was assessed as a logarithmic regression. Data were represented for each concentration as the mean ± standard deviation (SD) of the internal replicates. As the RTCA methodology retrieves non-dimensional readings, the collected data is consistent within each study, and different experiments should not be matched using the same reference frame or scale. Thus, the plots shown correspond to one representative independent experiment out of a total of three.

## 3. Results

### 3.1. Human Mesenchymal Stem Cells Exhibit Growth Inhibition Responses to SF_6_

Although subjected to the same experimental setup, the cell lineages tested in this study showed dissimilar responses to SF_6_ microbubbles, both in terms of susceptibility and growth over time. From all the conditions tested, assays using porcine fibroblasts mostly showed a dynamic profile (Figure 1A) characterized by a biphasic pattern: firstly, just after SF_6_ addition, an initial subtle decay was registered for normalized CI readings, followed by a steady increase spanning for the next 20 h; secondly, right after peaking, normalized CI readings decayed steadily until the end of the study (Figure 1A). It is worth noting that pinpoint spiked readings were recorded upon SF_6_ addition and were regarded as artifacts related to plate manipulation. In any case, for most of the conditions assayed, the registered normalized minimum cell index readings (CImin) never fell below 1, meaning that despite any registered decrease, the number of cells alive never fell below the number of cells present when SF_6_ was added. In the case of cultured porcine fibroblast control samples, normalized CI readings showed a behavior similar to that recorded for most assayed challenge conditions, characterized by the aforementioned biphasic pattern (Figure 1A). Notably, only in the assays inoculated with the highest SF_6_ dose, 308 μM, normalized CI readings experienced a quick and total decay, undoubtfully corresponding to significant cell death. To our surprise, assays exposed to high SF_6_ doses, except for the described case of 308 μM, performed relatively better than assays exposed to mid-range doses (Figure 1A).

Assays using hMSCs mostly developed a common pattern characterized by a mild normalized CI decrease, followed by somewhat stable readings for the rest of the study. As observed in the porcine fibroblast assays, short-lived impedance artifacts were briefly recorded in hMSC cultures upon challenge. In any case, shortly after SF_6_ inoculation, normalized CI readings decreased at every condition assayed (Figure 1B). Interestingly, that CI decrease was also recorded for the hMSC control samples that received saline devoid of SF_6_; however, these diverged in that CI readings recovered over time, a feature not registered for any other hMSC assay. Within the experimental assays, the impedance profile showed decreases ranging from 14.5% to 45.5% (TCI_min_), corresponding to 1.54 µM of SF_6_ at 16 h post challenge and 231 µM of SF_6_ 50 h post challenge, respectively (Figure 1B). Again, similarly to porcine fibroblasts, the higher SF_6_ concentration assayed, 308 µM, was highly deviated from the indicated norm, showing a quick and total decay of normalized CI readings characteristic of significant cell death (Figure 1B). Notably, for hMSCs, the 154 µM assay performance appeared to degrade over time. This was also recorded for the 231 µM dose, which further contrasted the others by a harsh CI decrease right after inoculation, followed by a partial restoration that peaked some 5 h after challenge.

Taken together, these observations strengthen the notion that interactions between suspended stem cells and SF_6_ microbubbles can promote cytotoxicity. Moreover, our data indicates that these responses are highly dependent not just on the microbubble concentration but, more importantly, on the specific cell line used.

### 3.2. Human Mesenchymal Stem Cell Responses Exhibit Dose-Dependent Behavior

Data collected for both cell lineages by means of RTCA was processed to determine any potential relation to the SF_6_ doses used and changes observed for impedance. While CImin readings allowed us to determine the toxicity/survivability at each assay, we also calculated the area under the curve (AUC) for every 10 h after challenge, which allowed us to understand the cumulative effects over time.

In the case of transgenic porcine fibroblasts, RTCA data processing was unsuccessful in finding any significant correlation for CI_min_ vs. SF_6_ concentration (i.e., r^2^ = 0.334, *p* = 0.130; Figure 2A) or for AUC vs. SF_6_ concentration (i.e., r^2^ < 0.42; Figure 2B) at any time point of all experiments carried out.

In the case of hMSCs, RTCA data processing allowed us to unveil significant parameters across conditions. Logarithmic correlation calculations corresponding to the curve containing data from 10 to 50 h after challenge and assessing survivability (CI_min_) vs. SF_6_ concentration provided significant r^2^ values, ranging between 0.898 and 0.811 (*p* < 0.001) (Figure 2C). Similarly, AUC correlation calculations corresponding to curves generated every 10 h in the post-challenge period up to the 50 h time mark provided relevant r^2^ values between 0.815 and 0.845 (Figure 2D). Notably, the linear regression analysis of maximum CI∇_%_ changes vs. SF_6_ concentration resulted in a highly significant correlation (r^2^ = 0.967, *p* < 0.001) when readings from 1.54 µM to 231 µM were used as input (Figure 2E).

Together, these analytical results support the notion that responses arising from direct SF_6_ microbubble interaction with cultured cells are highly dependent on the lineage involved. Moreover, this analysis indicates that highly susceptible lineages would very likely experience meaningful detrimental responses no matter the exposure to SF_6_ microbubbles, being either short-lived and/or of low dose.

## 4. Discussion

In this study, we provide, for the first time, observations on the potential cytotoxic consequences arising from the direct interaction between the ultrasonography contrast SF_6_ and cultured mesenchymal stem cells. By means of an RTCA system, we demonstrate the possibility of such effects on two distinct hMSC and *hCD55* transgenic porcine fibroblast cell lineages.

Ultrasonography procedures have multiple advantages over other imaging methods, including their relatively low cost and high availability, as well as the lack of ionizing radiation [16]. Nowadays, the intrinsic real-time dynamic imaging capabilities of ultrasonography techniques show increasing application thanks to the introduction of contrast-enhanced procedures. The safety of ultrasonography contrast agents has always been assessed considering the patient’s welfare, thus taking into account the nature of the indicated clinical procedures [17,18]. In this sense, SF_6_ is regarded as an inert and innocuous gas with poor solubility in aqueous solutions. In the case of SonoVue^®^, as well as others, proprietary phospholipid blends are used to stabilize SF_6_ into water-soluble micelle microbubbles that distribute into bodily fluids and degrade. When administered into any vascular compartment, about 80% of the gas is exhaled within 2 min after injection, reaching almost 100% within 15 min, indicating quick degradation dynamics within this environment. Seemingly, at the histological level, studies have shown that neither SF_6_ microbubbles nor low-mechanical-index ultrasound waves cause significant harm to vascular structures [19,20].

Nevertheless, cellular damage has been demonstrated in tissues exposed to SF_6_ formulations, especially when in conjunction with higher-mechanical-index ultrasound waves. More specifically, based on the high compressibility of SF_6_ microbubbles, which expand and collapse when exposed to ultrasound waves, this kind of setting allows for the sonication (sound-mediated integrity degradation) of newly formed blood vessels in tumors [6]. Moreover, this local degradation has been intentionally applied over the endothelial barrier as a useful tool to enhance targeted drug delivery in chemotherapy [7]. Elsewhere, SF_6_-mediated sonication has also proven useful for hyperlocal tissue degradation procedures, such as in the treatment of uterine fibroids [8].

Based on the above, considerable uncertainty arises in terms of direct cell–SF_6_ interaction having undesired effects on individual cells. In this sense, in clinical applications, independently of the mechanical index applied, SF_6_ microbubbles interact with biological structures as a whole, as the burden of any chemical or mechanical effect mediated by the exposure will be endured collectively by the cellular and extracellular structures in place. It is worth noting that the extracellular matrix in any given tissue is built to provide both mechanical support and protection. Distinctly, in the case of eco-guided cell transplantation procedures, only the naked cellular surface can withstand any interaction whatsoever, which poses a theoretical susceptible scenario for cell harm emerging from contact between the cell membrane and SF_6_ microbubbles. Such contact would promote subsequent micelle destabilization and cavitation, thus damaging the cell surface, as demonstrated in the references above [6,7,8]. In summary, this framework considers the notion that cell suspensions, devoid of the extracellular component and injected by means of contrast-enhanced eco-guided procedures, may be subject to a reduction in the number of viable cells administered.

Most methods used to study cytotoxicity are based on “release techniques”, in which chemical changes derived from cell membrane integrity disruption are detected [21,22,23]. Lesser common methods include flow cytometry, ELISA-mediated granzyme detection, radioactive labeling quantification, and morphometric microscopy analysis [24]. All these techniques are complex and vary greatly in precision. Most importantly, these methods tend to only provide single end-point measurements, requiring laborious and intricate setups in order to increase the number of conditions studied over time. In contrast, the RTCA xCELLigence^®^ System used in this study enables easy, dynamic, non-invasive, real-time monitoring of exposure-induced cytotoxicity, being devoid of any leverage for the assayed cells and operators. The only limitation of the RTCA methodology derives from the non-dimensional nature of its readings, nullifying the possibility of matching data from different studies in settings like ours. Nevertheless, the analysis package embodied in the system allows for thorough processing of data, providing better strength and scope to the conclusions when compared to other methods. Thus, from our perspective, the RTCA technology is the ideal system to assess the potential effects of SF_6_ on stem cells. To our knowledge, no other studies have been performed on the cytotoxic potential of the SF_6_ agent on any kind of stem-like cell lineage.

As evidenced from the Results Section, almost all conditions used to challenge porcine fibroblasts with SF_6_ microbubbles, including the controls, returned a biphasic impedance profile. This growth pattern is similar to that described in the literature for these cells in non-challenged RTCAs [12,14,25]. Notably, only the highest concentration assayed deviated from the norm. This behavior should not relate to medium degradation or cell starvation, since the amount of culture medium replaced with saline solution was common between the sham control and the highest concentration tested, with the former showing normal growth progression. On the contrary, the behavior shown by the 308 µM assays suggests that certain concentration thresholds should be surpassed in developing unmanageable cytotoxicity. In addition, partially deviating from that general trend, 231 µM assays showed an aggressive dip after challenge and a sluggish recovery afterwards (Figure 1A). This would mean that, in the case of porcine fibroblasts, exposure to SF_6_ microbubbles in concentrations ranging from 1.54 µM to 154 µM could be regarded as safe in terms of pleiotropic responses of suspended cells. Notably, we believe that the endurance shown by fibroblasts would make this lineage ideal for testing injection techniques requiring in-depth delivery, because in such settings, resorting to higher-end SF_6_ concentrations can help in obtaining better imaging and positional awareness.

The cytotoxicity of SF_6_ microbubbles was also found for the case of hMSCs, but this time, responses evolved in a wider range of concentrations (summary in Table 1). Similarly to *hCD55* porcine fibroblasts, most conditions assayed onto hMSCs behaved similarly in terms of CI profiles. Only the 308 µM, 231 µM, and 154 µM conditions deviated significantly in a negative manner from the common trend. This will add to the notion that a threshold exists in order to trigger unmanageable cytotoxic responses to SF_6_ microbubbles. Again, any effects derived from medium degradation could be disregarded, as hMSCs showed healthy dynamics for the complete duration of the study under the rest of the conditions. The notion of cell damage derived from direct surface interactions was also supported by logarithmic and AUC linear correlations results obtained for hMSCs, indicating cumulative effects depending not just on concentration but also time. Those observations were more potent in hMSCs due to the fact that, while being both mesenchymal lineages, fibroblasts are characterized as developing a rich extracellular matrix environment. As previously hinted, that extra physical layer may in fact dampen exposure from cavitation energy release, thus protecting fibroblasts from direct interaction with microbubbles. In contrast, hMSCs do not develop such a rich extracellular environment and lack such enhanced protection, thus showing a more progressive susceptibility profile. Therefore, in our test conditions, while both lineages are of mesenchymal origin, porcine fibroblasts can provide data representative of a more differentiated lineage compared to hMSCs.

Taking into account this study, we now retrospectively assess the potential cytotoxicity impact that the use of SF_6_ microbubbles may have had in prior experimental xenotransplantation procedures performed by our team. Based on our experience, in utero cell transplantation procedures do in fact constitute a vulnerable setting for the discussed matter. Nevertheless, in early research stages, this kind of setting, which implicates the delivery of stem cells into the developing fetus, has been widely proposed as a feasible method for specifically alleviating genetic-borne diseases, i.e., by providing cells producing factors or hormones missing in the patient [26,27,28]. Research in this field is still in the experimental stage, with most attempts in the literature resorting to complicated surgical procedures. Such techniques pose a significant risk of complications for both the pregnant individual and developing fetus. To overcome such limitations, in our group, we were able to inject hSMCs directly into pig fetuses without further surgical procedures, thus simplifying the process [10]. In such a procedure, suspended cells had to be injected into the fetus peritoneal cavity for the possibility of effective long-term implantation. This requisite made for an intricate procedure in which resorting to contrast-enhanced imaging greatly helped for on-target delivery. Overall, according to our experience, the eco-guided in utero cell transplantation technique allowed for accurate delivery while keeping pregnancy complications and newborn piglet survival rates within normal range. Yet, in the study, the number of piglets retaining human cells at birth was below expectations [10].

The details above caused us to consider whether variations in cell implantation could arise from the use of SF_6_ contrast. This possibility was not considered at the time nor could be identified retrospectively, thus providing reason for the current paper. It is worth considering that, in contrast to the vascular compartment, the peritoneal cavity of developing piglets is devoid of any significant mechanical clearance. Consequently, in this specific case, SF_6_ microbubbles would remain in the cavity with the cells until total degradation, which justifies our extended assay times, being roughly 2 days (50 h). Now, we suggest considering hSMC depletion by the formula extrapolated by comparing maximum CI∇_%_ vs. the final concentration of saline-resuspended SF_6_ microbubbles (Figure 2E). In our work on hSMCs in in utero cell transplantation to obtain chimeric piglets [10], injected cells ranged from 2 × 10^6^ to 15 × 10^6^ in a total volume of 1 mL, comprising 0.5 mL of PBS loading cells and another 0.5 mL of reconstituted SonoVue^®^ solution. This accounts for a 1:2 final dilution for the contrast agent, corresponding to the 1.54 µM assay in this study. Since hMSCs under this condition experienced a 14.5% decrease in CI, by applying this correction to inoculated inputs, the effective doses transferred ranged from 1.7 × 10^6^ to 12.75 × 10^6^ cells. Interestingly, when considering the cell load injected in our previous study, success rates, which were determined by detecting transgenic cells from piglet blood samples by means of flow cytometry, were considerably higher when lower cell loads were used (2 × 10^6^ cells 31%; 3.5 × 10^6^ cells 69%; 10.75 × 10^6^ cells 0%; 15 × 10^6^ cells 0%). This could be related to increased microbubble instability derived from cramped conditions developing in the injection solution, as in a higher-cell-density environment, the possibility of cell membrane damage from cavitation would grow exponentially.

The methodology proposed here has potential to help fine-tune settings in contrast-enhanced eco-guided delivery in cell therapy procedures. In the case of our group activity, efficiently adjusting the required number of cells to be administered while keeping highly precise delivery characteristics was crucial, as both features strongly impact the overall complexity, performance, and cost of experimental procedures. Such development becomes especially useful for the obtention of chimeric animals by in utero transplantation. It is also important to keep in mind that the correlation between the maximum CI∇_%_ vs. the final SF_6_ concentration also accounts for the effects from accumulated exposure, as maximum CI∇_%_ values occurred at different times depending on concentrations. In the case of in utero injections, as hinted above, this becomes relevant due to the absence of any mechanical clearance of the peritoneal cavity. Nevertheless, beyond providing improvements for this specific type of xenotransplantation research, we are confident that our findings would also be of interest to clinicians practicing analogous procedures, as we demonstrate how effects may be found even in very short-lived exposures at low concentrations depending on cell type. In this sense, as hinted by the data in our previous work, when dealing with especially delicate cell lineages, properly adjusting SF_6_ contrast to cell ratios could be the difference between failure and success.

We considered if SF_6_ contrast could be used for eco-guided cell transplantation and determined a suitable methodology to properly adjust the number of cells to inject, with potential benefits to patients, in the form of more reliable procedures and overall procedure performance improvement in terms of protocol optimization and cost management. Future testing on induced pluripotent stem cells (iPSCs) as well as neural progenitors should be considered for their potential therapeutic value.

## 5. Conclusions

Taken together, the evidence collected in this study supports the use of SF_6_ as a convenient agent for contrast-enhanced eco-guided cell delivery in clinical procedures. However, the different responses observed between hMSCs and porcine fibroblasts indicate that SF_6_ microbubbles, despite their clinical safety, should not be considered harmless to the cells administered. Thus, we strongly suggest testing in advance phospholipid-stabilized gas microbubble ultrasonographic contrast agents along with specific cell lineages to determine potential effects on cell viability, thus determining the impact on technical performance and the reproducibility of procedures.

## Figures and Tables

**Figure 1 biotech-14-00086-f001:**
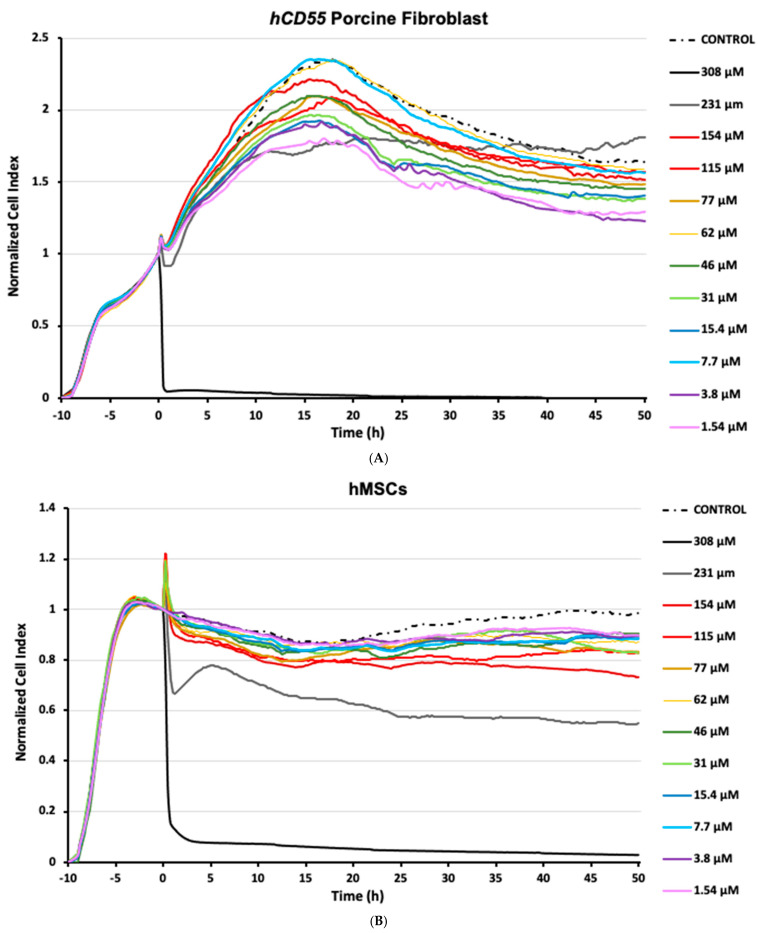
Impedance-based label-free monitoring of cells exposed to SF_6_. Cells were allotted 10 h of growth time after seeding and before challenge. The normalized cell index (CI) is a dimensionless parameter generated from impedance variations over time depending on culture confluence evolution. Tested SF_6_ conditions are shown in color by concentration (µM). Sham control assays were exposed to saline carriers corresponding to the highest assayed concentration. (**A**) CI readings obtained for *hCD55* transgenic porcine fibroblasts. (**B**) CI readings obtained for human mesenchymal stem cells (hMSCs). Tracking plots are representative of three separate independent studies.

**Figure 2 biotech-14-00086-f002:**
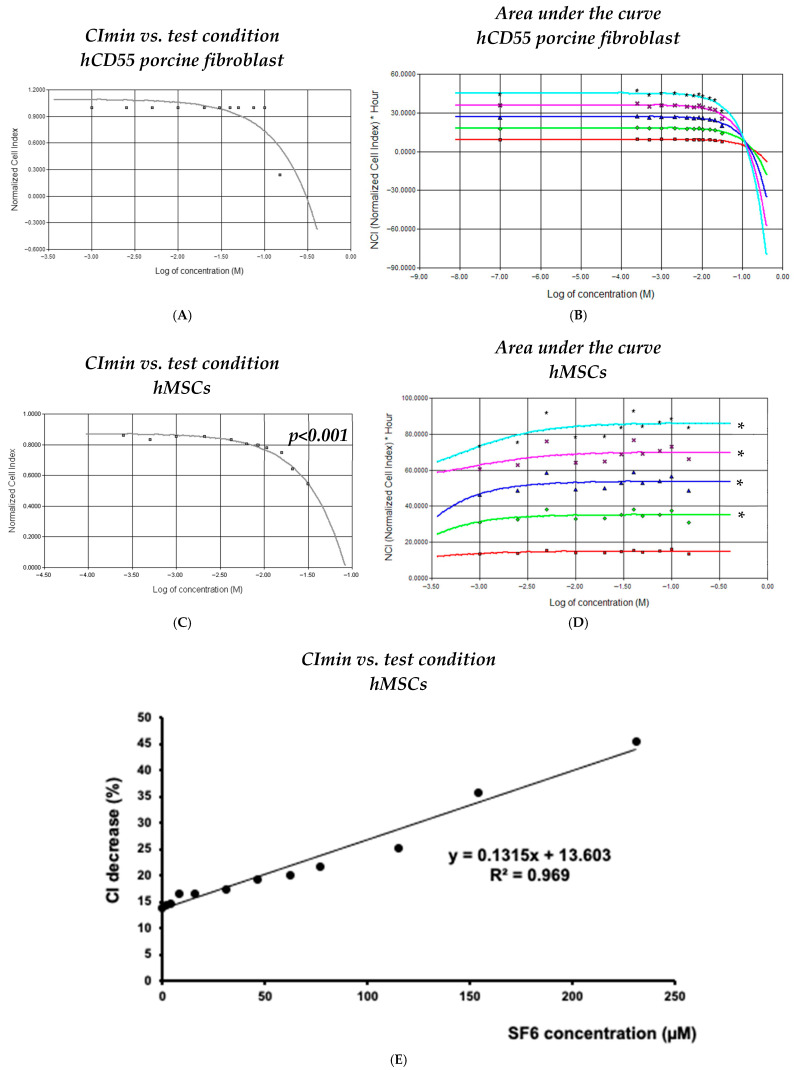
Dynamic analysis of responses obtained from cell exposure to SF_6_. (**A**,**C**) Normalized minimum cell index (CI) readings recorded for each assayed cell line and SF_6_ microbubble concentration; (**B**,**D**) area under the curve, representative of cumulative effects calculated at 10 h (red), 20 h (green), 30 h (blue), 40 h (magenta), and 50 h (cyan) after SF_6_ inoculation. (**E**) Logarithmic regression obtained for the detected percentage CI decrease vs. the final volume fraction of SF6 microbubbles resuspended in saline; the decline in cell viability correlates with SF_6_ microbubble concentration. The plots shown are representative of three different experiments. Asterisks denote statistical relevance (*p* < 0.001).

**Table 1 biotech-14-00086-t001:** Outcomes of SF_6_ exposure on human mesenchymal stem cells (hMSCs).

Effect vs. SF_6_ [µM]	1.54	3.8	7.7	15.4	31	46	62	77	115	154	231	308
Sharp initial CI decrease										x	x	x
Early deviation from sham							x	x	x			
Late deviation from sham	x	x	x	x	x	x						
Cumulative (*p* < 0.001)	x	x	x	x	x	x	x	x	x	x	x	x
Maximum CI decrease (%)	14.48	14.58	16.62	16.68	17.44	19.3	20.09	21.76	25.13	35.77	45.55	84.31

## Data Availability

The original contributions presented in this study are included in the article. Further inquiries can be directed to the corresponding author.

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
