# Peer review of "Exposure to Sulfur Hexafluoride Influences Viability in Cell Transplant Suspensions"

_biotech, 2025, doi:10.3390/biotech14040086_

Round 1
Reviewer 1 Report
Comments and Suggestions for Authors
Cell therapy is a promising approach for replenishing the lost tissue-specific cells and/or protecting the damaged tissue from secondary injury. On the other hand, Sulphur-hexafluoride (SF6) microbubbles has been used as the contrast agents for sonographic imaging. Authors asked the potential cytotoxicity of this agent when it is used as the guide for cell-injection. Using real-time cell analysis (RTCA), the authors provide valuable in vitro data demonstrating differential effects of SF₆ exposure on two mesenchymal lineages: human mesenchymal stem cells (hMSCs) and transgenic porcine fibroblasts. The topic is novel, clinically relevant, and methodologically reliable. Overall, the study holds considerable promise for improving the precision and efficiency of cell-based therapeutic procedures. However, there are several issues that should be addressed to enhance the clarity, impact, and rigor of the manuscript.
Major Concerns:
1. Generalizability of cell models
The study includes only two cell types (hMSCs and porcine fibroblasts). While this is acceptable for a preliminary investigation, the manuscript should acknowledge the limited generalizability and suggest future testing with additional therapeutically relevant cell types (e.g., iPSCs, neural progenitors).
2. Background clarity and depth
While the introduction outlines the relevance of SF6, the mechanistic basis of its potential cytotoxicity, particularly the biophysical interactions between microbubbles and cellular membranes, would be better to be described in greater detail, with citing appropriate literatures.
3. In vitro–in vivo extrapolation
Although the manuscript attempts to link the findings to prior in vivo transplantation work, the connection remains mostly speculative. A clearer and quantitative comparison between predicted cell loss (based on RTCA) and observed engraftment inefficiencies in vivo would strengthen the translational claim.
4. Redundancy in discussion
Several arguments, particularly those related to the protective role of the extracellular matrix and the biphasic growth pattern of fibroblasts, are repeated throughout the discussion (lines 225-230, 260-265, 280-285), often without additional mechanistic or comparative elaboration. A more concise and focused narrative would improve readability.
5. Data presentation
Figures 1 and 2 would benefit from more detailed labeling (e.g., units, CImin definition). In addition, the legends could be made more self-contained.
Minor comments:
a. There are several spelling or grammatical errors listed below:
a-1. Line 23 "has is" might be "have".
a-2. Line 63 'util' should be 'until'.
a-3. Line 136 "308 µM mM" should be "308 µM".
a-4. Line 197 'strength' should be 'strengthen'.
a-5. Line 340 "his time" should be "this time".
a-6. Line 359 "Taking in account" should be "Taking into account".
Author Response
Dear Reviewer 1,
We are greatful for your feedback and suggestions. We think that your input helps in making a more consistent paper. Please, find below detailed responses to your comments and considerations. We hope the amends integrated fulfill your requests for publication. Thanks for your support.
S. Liarte on behalf of the team.
DETAILED RESPONSE - Review comments displayed a non bold characters
1. Generalizability of cell models . The study includes only two cell types (hMSCs and porcine fibroblasts). While this is acceptable for a preliminary investigation, the manuscript should acknowledge the limited generalizability and suggest future testing with additional therapeutically relevant cell types (e.g., iPSCs, neural progenitors).
RESPONSE - Dear Reviewer, thank you for pointing this. Indeed, this limitation was already implicit in the text we originally submitted, as it can be found in the abstract in line 32, at the end of the discussion’s last paragraph and in conclusions in lines 416-417-418. Nevertheless, we think that hinting specific lineages as you suggested is a good thing, so in the current version of the manuscript we included a sentence making specific reference to iPSCs and neural progenitors at the end of the discussion.
2. Background clarity and depth While the introduction outlines the relevance of SF6, the mechanistic basis of its potential cytotoxicity, particularly the biophysical interactions between microbubbles and cellular membranes, would be better to be described in greater detail, with citing appropriate literatures.
RESPONSE - Thanks for indicating this. We included new text at the second paragraph on the introduction further providing details on this. We hope this suffices your request.
3. In vitro–in vivo extrapolation Although the manuscript attempts to link the findings to prior in vivo transplantation work, the connection remains mostly speculative. A clearer and quantitative comparison between predicted cell loss (based on RTCA) and observed engraftment inefficiencies in vivo would strengthen the translational claim.
RESPONSE- Thank you for the suggestion, that was a really good point. A new piece of text in the discussion tries to better illustrate such relation. From our previous work, although no different concentrations of SF6 were used, the varying number of cells injected provides a valid surrogate for that relation, as high cell density injections, thus with increased probability of damage by cavitation, performed clearly worst than diluted ones.
4. Redundancy in discussion Several arguments, particularly those related to the protective role of the extracellular matrix and the biphasic growth pattern of fibroblasts, are repeated throughout the discussion (lines 225-230, 260-265, 280-285), often without additional mechanistic or comparative elaboration. A more concise and focused narrative would improve readability.
RESPONSE - We went through the text and we do not agree with this point. While we may reiterate a few times, this is done on purpose in other to strengthen the idea that is being discussed. We believe that most readers would appreciate this. We hope that the reviewer may understand this.
5. Data presentation Figures 1 and 2 would benefit from more detailed labeling (e.g., units, CImin definition). In addition, the legends could be made more self-contained.
RESPONSE - We performed improvements following directions. Now we clarify in legends that CI is a normalized dimension-less parameter. This is the way the RTCA system deals for plotting impedance profiles generated from distinct cell lines.
Minor comments: a. There are several spelling or grammatical errors listed below:
a-1. Line 23 "has is" might be "have". CORRECTED
a-2. Line 63 'util' should be 'until'. CORRECTED
a-3. Line 136 "308 µM mM" should be "308 µM". CORRECTED
a-4. Line 197 'strength' should be 'strengthen'. CORRECTED
a-5. Line 340 "his time" should be "this time". CORRECTED
a-6. Line 359 "Taking in account" should be "Taking into account". CORRECTED
Reviewer 2 Report
Comments and Suggestions for Authors
The research examines the potential cytotoxic impact of the ultrasonographic contrast agent sulphur hexafluoride (SF6) on two cell types — human mesenchymal stem cells (hMSCs) and transgenic porcine fibroblasts — utilizing a real-time impedance monitoring assay (RTCA). This investigation addresses a timely and clinically significant question, as SF6 is commonly employed for contrast-enhanced, ultrasound-guided cell delivery, yet its influence on cell survival is not well characterized. While the study is methodologically robust and well organized, several aspects need further clarification and refinement to enhance its overall scientific value.
1. Figures 1 and 2 in the manuscript lack some important presentation details:
-
Both figures have basic legends but do not specify axis units (e.g., CI is shown but not fully explained on axes).
-
They lack statistical markers (p-values, significance indicators) despite text mentioning significant correlations (r² and p-values).
-
Legends are brief and do not fully explain each curve and symbol, requiring readers to infer details from the text.
For example:
Figure 1 shows “normalized cell index readings” for porcine fibroblasts and hMSCs but does not clearly indicate the SF₆ concentrations tested on the axes.
Figure 2 breaks down results by CImin, AUC, and CI% decrease but lacks clear unit annotations and statistical labels (only regression r² is referenced in the text)2.
2. The manuscript only says experiments were done “three times, with duplicates for each condition” .
-
-
-
It does not clearly state whether these were biological or technical replicates, nor does it give a full description of statistical tests (p-values are mentioned in some figure analyses, but not systematically across results).
-
-
3. A summary table detailing experimental parameters—including SF6 concentrations and key outcomes (CImin, AUC, percentage CI reduction) would make the results easier to interpret without examining multiple graphs.
4. It is unclear whether control cells (unexposed to SF6) underwent the same handling as treated cells, aside from the exposure itself.
5. The discussion dedicates considerable space to general information about SF6 but provides limited insight into underlying mechanisms, such as the potential role of microbubble–cell membrane interactions in driving cytotoxicity.
6. The authors should establish a stronger connection to clinical applications, including how the findings could inform cell dose adjustments or strategies to minimize SF6-related cell damage.
Author Response
Dear Reviewer,
We are greatful for your feedback and suggestions. We think that your input helps in making a more consistent paper. Please, find below detailed responses to your comments and considerations. We hope the amends integrated fulfill your requests for publication. Thanks for your support.
S. Liarte on behalf of the team.
DETAILED RESPONSE - Review comments displayed a non bold characters
1. Figures 1 and 2 in the manuscript lack some important presentation details:
• Both figures have basic legends but do not specify axis units (e.g., CI is shown but not fully explained on axes).
⁃ RESPONSE - Now we clarify in legends that CI is a normalized dimension-less parameter.
• They lack statistical markers (p-values, significance indicators) despite text mentioning significant correlations (r² and p-values).
⁃ RESPONSE - Significance indicators had been added to figure 2. Such thing is not possible for figure 1, this is derived from the RTCA methodology. As it provides dimension-less readings, collected data is consistent just within each study. Trends can be compared at glance. This was described in section 2.6.
• Legends are brief and do not fully explain each curve and symbol, requiring readers to infer details from the text.
⁃ RESPONSE - We tried to improve legends. We hope it suffices your request.
For example:
Figure 1 shows “normalized cell index readings” for porcine fibroblasts and hMSCs but does not clearly indicate the SF₆ concentrations tested on the axes.
RESPONSE - Concentrations assayed are indicated by color, this is now further clarified in legends.
Figure 2 breaks down results by CImin, AUC, and CI% decrease but lacks clear unit annotations and statistical labels (only regression r² is referenced in the text)2.
RESPONSE - We further clarified legends and added statistical significance labelling.
2. The manuscript only says experiments were done “three times, with duplicates for each condition” .
▪ It does not clearly state whether these were biological or technical replicates, nor does it give a full description of statistical tests (p-values are mentioned in some figure analyses, but not systematically across results).
⁃ RESPONSE - text has been amended to better illustrate that replicates were of the biological kind. A mention to p-values calculation has been included in section 2.6.
3. A summary table detailing experimental parameters—including SF6 concentrations and key outcomes (CImin, AUC, percentage CI reduction) would make the results easier to interpret without examining multiple graphs.
RESPONSE - This is a great suggestion. A table was generated following indications.
4. It is unclear whether control cells (unexposed to SF6) underwent the same handling as treated cells, aside from the exposure itself.
RESPONSE - This was already indicated in the methods section, however now we include in the legend of figure 1 the expression “sham controls” to better illustrate that handling was the same.
5. The discussion dedicates considerable space to general information about SF6 but provides limited insight into underlying mechanisms, such as the potential role of microbubble–cell membrane interactions in driving cytotoxicity.
RESPONSE - This has been detailed in the introduction section as per request of a different reviewer.
6. The authors should establish a stronger connection to clinical applications, including how the findings could inform cell dose adjustments or strategies to minimize SF6-related cell damage.
RESPONSE - The last paragraphs at the discussion section had been modified to better ilustrate bennefits. We hope this suffices your request.
Round 2
Reviewer 1 Report
Comments and Suggestions for Authors
Authors addressed my previous review comments and the manuscript is now ready for the publication.